# Mathematical Approach to Estimating the Main Epidemiological Parameters of African Swine Fever in Wild Boar

**DOI:** 10.3390/vaccines8030521

**Published:** 2020-09-12

**Authors:** Federica Loi, Stefano Cappai, Alberto Laddomada, Francesco Feliziani, Annalisa Oggiano, Giulia Franzoni, Sandro Rolesu, Vittorio Guberti

**Affiliations:** 1OEVR—Sardinian Regional Veterinary Epidemiological Observatory, Istituto Zooprofilattico Sperimentale della, Sardegna, 09125 Cagliari, Italy; stefano.cappai@izs-sardegna.it (S.C.); sandro.rolesu@izs-sardegna.it (S.R.); 2Department of Animal Health, Istituto Zooprofilattico Sperimentale della Sardegna, 07100 Sassari, Italy; alberto.laddomada@izs-sardegna.it (A.L.); annalisa.oggiano@izs-sardegna.it (A.O.); giulia.franzoni@izs-sardegna.it (G.F.); 3Italian Reference Laboratory for Pestivirus and Asfivirus, Istituto Zooprofilattico Sperimentale dell’Umbria e delle Marche “Togo Rosati”, 06126 Perugia, Italy; f.feliziani@izsum.it; 4Department of Wildlife (Ozzano Emilia), ISPRA—Superior Institute for Environmental Protection and Research, 00144 Rome, Italy; vittorio.guberti@isprambiente.it

**Keywords:** african swine fever, epidemiology, basic reproduction number, force of infection, disease transmission, wild boar, Sardinia

## Abstract

African swine fever (ASF) severely threatens the swine industry worldwide, given its spread and the absence of an available licensed vaccine, and has caused severe economic losses. Its persistence in wild boar (WB), longer than in domestic pig farms, and the knowledge gaps in ASF epidemiology hinder ASF virus (ASFV) eradication. Even in areas where disease is effectively controlled and ASFV is no longer detected, declaring eradication is difficult as seropositive WBs may still be detected. The aim of this work was to estimate the main ASF epidemiological parameters specific for the north of Sardinia, Italy. The estimated basic (*R*_0_) and effective (*R*_e_) reproduction numbers demonstrate that the ASF epidemic is declining and under control with an *R*_0_ of 1.139 (95% confidence interval (CI) = 1.123–1.153) and *R*_e_ of 0.802 (95% CI = 0.612–0.992). In the last phases of an epidemic, these estimates are crucial tools for identifying the intensity of interventions required to definitively eradicate the disease. This approach is useful to understand if and when the detection of residual seropositive WB is no longer associated with any further ASFV circulation.

## 1. Introduction

African swine fever (ASF) threatens the swine industry worldwide due to its epidemiological characteristics and its current spread in both domestic and wild pig populations. ASF has been defined as the most serious animal disease that the world has had for a long time, if not ever [1]. The spread of the etiological agent (African swine fever virus (ASFV)) is having negative repercussions on animal health and animal welfare at an international level [2,3,4]. The disease’s ability to affect both domestic pigs (*Sus scrofa domesticus*) and wild boar in Europe, and *Phacoerus africanus* or *Potamochoerus* spp. in Africa, makes its control and eradication even more difficult [5,6,7]. ASFV has different, recognized epidemiological reservoirs such as wild boar in most of the European countries (i.e., Baltic countries), backyard pig farms with low biosecurity (i.e., Danube Delta and south and eastern Africa), or illegal free-ranging pigs (i.e., Sardinia, Italy) [8,9,10,11]. In the European Union (EU), considering the absence of a licensed vaccine or treatment against ASF, to date, the available control measures focus on preventing and controlling the spread of the virus by increasing biosecurity measures, the “stamping out” for domestic pigs within infected farms, and the isolation of the wild boar-infected areas managed by active and passive surveillance (EU Council Directive 2002/60/EC of 27 June 2002, and EC Guidelines for ASF strategic approach) [12].

Measures to avoid and prevent infected pigs or pork products have been introduced into ASF-free regions [2,13,14]. If these disease control measures provide the tools for ASFV control in the EU, the extension of these restrictive measures beyond necessary could negatively affect the food chain economy. Given the complexity of the measures used to control and eradicate the disease, the final procedure to regain freedom from the disease could be even more complicated when wild populations are involved due to their dynamic nature, the sanitary checks often not being based on calculated sample size, and intensity, time, and space not being previously defined. Consequently, a defined, robust, and shared exit strategy is lacking for when, in absence of ASFV evidence, only a few and almost sporadic cases of seropositive wild boar are detected [15,16].

In some European countries, such as part of Estonia and Latvia, and Sardinia (Italy), ASF notifications are now limited to seropositive PCR-negative wild boar individuals [17,18,19,20,21]. Although the final eradication of an infectious disease in wildlife is always difficult to assess [22], data on the prevalence of infection (i.e., virologic and serological data) can be used to estimate some epidemiological parameters to more accurately describe the whole epidemiological evolution of the infection in a specific area/habitat [23,24,25,26]. Particularly, the estimates of the reproduction number *R* associated with the epidemic provides a measure of the intensity of interventions required to achieve control and finally eradicate the disease [9,27,28,29].

In this work, we aimed to confirm the ancillary role played by wild boar in the epidemiology of ASF in Sardinia, Italy, where the eradication strategy is based on the epidemiological assumption that the local, illegal, free-ranging pig population is the true ASFV reservoir and, once successfully removed, ASF would fade out almost spontaneously in the wild boar population [11].

Since the most likely hypothesis assumes that different trends in virus circulation in wild boar could be associated with animal density, type of land, presence of free-ranging pigs living in contact with wild boars, and compliance of hunting companies with the rules of disease management, we considered the wild boar management unit (WB-MU) of the Anglona-Gallura area. This area is characterized by the absence of free-ranging pigs, a relatively known and low-density wild boar population (average of 3.2, SD of 0.34 wild boars/km^2^ versus average of 5.2, SD of 0.62), the prevalent type of land as Mediterranean scrub, and with a very high level of compliance with the rules of the last ASF Eradication Plan (ASF-EP15-18), over 85% (mean = 85.8, 95% confidence interval (CI) of the mean = 73.4–98.2). For this purpose, the epidemiological situation of the north of the island during the last 10 years was studied in depth and an optimization analysis based on a stochastic model was conducted to validate the estimated epidemiological parameters.

## 2. Materials and Methods

### 2.1. Sardinian Epidemiological Context

As described in several studies, the epidemiological cycle in Sardinia involved three different swine populations: Domestic pigs (in farms), sylvatic wild boar, and illegal, free-ranging pigs. The latter are regarded as the main ASFV reservoir on the island [30,31]. Efficient depopulation actions against free-ranging pigs were implemented and about 4500 illegal pigs were culled between December 2017 and February 2020 [11]. The disease completely disappeared from domestic pigs in 2018 due to the strengthening of control measures, increasing biosecurity, and the compliance with the rules of the ASF-EP15-18 (Figure 1A) [30]. Although the disease prevalence in wild boar reached high levels, from 2013 the PCR-positive and seropositive trends recorded a strong decrease that was even more evident from 2015 onward, as illustrated in Figure 1B.

Particularly, in Sardinia, most of the ASF cases occur from February to June, with peak incidence in May and lowest disease activity in December (Figure 2a). In domestic pig farms, the virus is most active in May (26%) and June (21.2%), during which about half of the outbreaks occur (Figure 2b). ASF seasonality in wild boar, considering only passive surveillance, peaks in terms of number of virus positive cases from February to May, with a peak in March when about 40% of ASFV-positive cases occur (Figure 2c). This suggests seasonality in contact rate, evenly caused by the reproduction period at the end of hunting season, as underlined in several studies [32,33].

The hunting season, running from November to January, is managed by 11 WB-MUs (Figure 3) with different control measures applied inside and outside the infected zone (IZ), as established by ASF-EP15–18 (and subsequent additions). Four WB-MUs are totally or partially included inside the IZ and different disease trends are associated with each one: Anglona-Gallura, Nuoro-Baronia, Goceano-Gallura, and Gennargentu-Ogliastra (Figure 4).

### 2.2. Wild Boar Management Unit of Anglona-Gallura

This study was based on ASF events in wild species originated from the WB-MU of Anglona-Gallura, located in the north of the island (Figure 3). The Anglona-Gallura territory includes 45 municipalities and covers a total of 3040 km^2^, bordered by the sea; its territory is mainly characterized by hilly small plateaus of a volcanic or limestone nature. The estimates of the wild boar population average in Anglona-Gallura is approximately 13,550 animals, based on the wildlife management plan and as explained in previous studies [21,30,34]. The presence of free-ranging pigs, with notoriously high interaction rate with wild boar population in Sardinia [11,30,31], has never been detected within this area.

The hunting season in Anglona-Gallura is limited during November–January and includes eight hunting days/by month (Week 1a, Week 1b, Week 2a, Week 2b, Week 3a, Week 3b, Week 4a, Week 4b), corresponding to each Sunday (a) and Thursday (b) [30]. Inside the IZ, the samples from each hunted animal must be delivered to the veterinary services to be tested by the ASF Laboratory, Istituto Zooprofilattico Sperimentale della Sardegna (IZS-Sardegna). Contemporarily, the carcasses of hunted animals must be stored until the ASF tests results are received, and are destroyed if positive or delivered if negative. Data including sex and age of these animals, as well as hunting company, hunting day, and geolocation are collected by the veterinary services. ASF control during the remaining period is based on passive surveillance, which is limited to the few animals found dead reported by competent authorities, and samples are tested by IZS-Sardegna. Data regarding the age, sex, and geolocation of these samples are not recorded.

### 2.3. Data Collection

Wild boar sample were collected from both active (hunted animals) and passive surveillance (found-dead animals). Serum samples were tested for ASFV antibodies, and organs (mainly spleens) were assayed for the ASFV genome. ASFV antibodies’ presence was first assessed using a commercial ELISA test (Ingezim PPA Compac^®^, Ingenasa, Madrid, Spain), then positivity was confirmed through immunoblotting (IB) [15]. Serum samples were considered positive when they scored positive in both the screening (ELISA) and the confirmatory (IB) tests. Presence of ASF viral genome was assessed by real-time PCR [15,35].

For the purposes of this work, an ad hoc database was created for the study period of 2011–2020 (Table 1). Data regarding ASF outbreaks in wild boar were retrieved from the Italian National Informative System for Animal Disease Notification (SIMAN). Specific data about hunting location, data of disease suspect, age of the animal tested (young, 0–6 months; subadult, 6–18 months; and adult, >18 months) and results of ASF laboratory tests were collected for the internal laboratory database of IZS-Sardegna (SIGLA). Any contemporary presence of both antigen and antibody for ASFV is reported in Appendix A. Data quality was assessed in terms of accuracy, completeness, and missing information. The ASF epidemic in Anglona-Gallura began on 20 September 2011 on a domestic pig farm in the Monti municipality and gradually disappeared due to the strongly implemented control measures. In the same WB-MU, the first ASF case was found on 22 January 2012 in an adult wild boar during the 2011–2012 hunting season, during which a total of 119 wild boars were tested for ASFV and 967 for antibody presence (Table 1). During the next hunting seasons (2012–2020), a total of 6257 and 14,337 wild boar were virologically and serologically tested, respectively. Overall, six PCR-positive tests were detected in subadult animals and 18 in adult, whereas the presence of ASFV antibodies still persists in seropositive animals, showing a decreasing trend of the disease over the period (Figure 4). The last PCR-positive case in Anglona-Gallura dates back to 26 December 2015 in Tergu. As shown in Appendix A, during the 2013–2014 hunting season, a total of 3 (0.6%) hunted wild boar that were PCR-positive also presented antibodies for ASF; in two of these animals, ASFV was isolated using the Malmquist test. The percentage of contemporary PCR- and antibody-positive wild boar decreased to 0.4% during 2015–2016 and was 0% in the following years. ASF positivity has never been detected by passive surveillance within this area (Appendix A). Given the low number of samples from passive surveillance and the problem related to its usage (i.e., low samples covering different time periods), the parameter calculation only considered data from active surveillance.

In Anglona-Gallura, a total average of 19 animals by hunting day were hunted and tested for ASFV genome and antibody presence. Table 2 exhibits the hunting-day mean, standard error of the mean (SEM), 95% CI of the mean, and width of the CI for the observed data. The number of wild boar hunted on Sunday (Weeks a) was consistently higher (about four times more) compared to Thursday (Weeks b). Weeks 2a and 2b were the hunting days with the highest number of wild boar (about 42 and 40 animals, respectively). Week 1b recorded the lowest mean value (mean = 8.06, SEM = 0.35).

As shown in Figure 5, during the first hunting season (2011–2012), only one adult PCR-positive wild boar was detected. During the next season, one adult PCR-positive and four seropositive (one young, one subadult, and two adult) animals were detected. In the 2013–2014 hunting season, a total of six subadult and 10 adult PCR-positive, and three young, five subadult, and six adult seropositive wild boar was detected. Subsequently, the number of ASFV detected in subadult animals decreased to zero, while adult virus-positive animals persisted for two more years. During the last three years, only adult ASF seropositive animals were found.

### 2.4. Estimates of the ASFV Force of Infection

The force of infection (*λ*) parameter describes the risk that a susceptible individual is newly infected between *t* and *t* + 1 times. Given its definition, *λ* strictly depends on the number of infectious individuals and the rate at which the susceptible individuals came into “effective contact” with those infectious. Three methods are used to estimate the force of infection.

#### 2.4.1. Method 1: Estimating λ by Contact Rate

Supposing that the number of infectious at time *t* (*I*(*t*)) is represented by the number of animals tested as virus positives and the parameter *β* describes the average contact rate, the force of infection can be estimated by the Equation (1) [24,26,36]:(1)λ(t)=β×∑I(t)
where time *t* was defined as the six months of 2011–2012 and 2012–2013 hunting seasons. For this parameterization, the average rate of contact between animals (*β*), reported by Bosch et al. as being 0.5/day, was applied. The final daily *λ* was obtained by dividing by approximately 180 days the second term of the equation (*I*(*t*)).

#### 2.4.2. Method 2: Estimating λ Using a Catalytic Model

Assuming that contacts between individuals are mixed randomly, it is possible to apply a catalytic model to estimate *λ* from serological data by age class [37,38]. Assuming that individuals are infected at a constant annual rate independent of age and calendar year, the rates of change in the proportion of susceptible individuals, *s*(*a*), and those ever infected, *z*(*a*), with respect to age *a*, are given by:(2)$$\{\begin{array}{c} \frac{ds\left(a\right)}{da}=-\lambda s\left(a\right)\;\mathrm{where}\;s\left(a\right)=e^{-\lambda a}\\ \frac{dz\left(a\right)}{da}=\lambda s\left(a\right)\;\mathrm{where}\;z\left(a\right)=1-e^{-\lambda a} \end{array}$$

Considering the age classes of the hunted animals, data from the laboratory test results of 2011–2012 and 2012–2013 hunting seasons were used to implement the age dependent-based catalytic model on Equation (2), generating a monthly *λ* value. The goodness of fit of this model and the final value of *λ* were evaluated using the maximum likelihood method.

#### 2.4.3. Method 3: Estimating λ Based on Proportion of Infected

Assuming that the force of infection is not age dependent and individuals mix randomly, this parameter can be calculated without a mathematical model but instead by applying parametric calculations, defining *λ* as the reciprocal of average age at infection (*A*) [36], which can be obtained as:(3)$$\lambda =\frac{1}{A}\approx \frac{1}{s\times L}$$
where *L* is the average life expectancy, established as 30 months (considering the main age of wild boar hunted in Sardinia), and *s* denotes the proportion of susceptible population. Basically, assuming that the proportion of individuals who test negative for infection (i.e., PCR-negative and antibody (Ab)-negative) is equal to the proportion of susceptible, thus *s* can be estimated based on individual age as:(4)$$s_{t}=\underset{t,a}{\sum }\pi_{a}\frac{S_{t,a}}{N_{t,a}}$$
where π_a_ is the proportion of the population in the age group *a*, *S_t_*_,*a*_ is the number of susceptible (ASF-tested negative) among those tested in age group *a* at time (*t*), and *N_t_*_,*a*_ is the number of animals of each age group *a* that were ASF-tested [26]. The proportions *a* of wild boar population in each age group were defined in the literature as 0.63, 0.19, and 0.18 for young, subadult, and adult, respectively, considering intermediate environmental conditions [39,40]. This calculation was applied using data from the 2011–2012 and 2012–2013 hunting seasons. Subsequently, to generate a time-unit measure of one day for *λ*, the result was divided for about 30 days (1 month).

### 2.5. Basic Reproduction Number (*R*_0_) Estimation

The basic reproduction number (*R*_0_) is the average of secondary infectious individuals that would result when one infectious individual is introduced for the first time into a completely susceptible population [41]. This parameter is able to quantify the spread of an infectious disease, predicting its speed, scale, and the level of herd immunity required to contain the disease. The higher the value of *R*_0_, the faster the epidemic progresses [36]. Assuming a well-mixed population, when a proportion *p* of the population is effectively protected from infection at time *t* (for *t* > 0), this parameter is known as the effective reproduction number (*R_e_*), which is related to *R*_0_ by *R_e_ =* (1–*p*) × *R*_0_ [24]. When *R_e_* ≤ 1, the epidemic is in decline and may be regarded as being under control at time *t* (and vice versa when *R_e_* > 1).

Although several calculations for *R*_0_ have been proposed, in this work, four different methods were applied based on the main assumption that individuals mix randomly in an endemic context.

#### 2.5.1. Method 1: Estimating *R*_0_ from the Doubling Time

The basic reproduction number was calculated based on the epidemic doubling time method proposed by Barongo et al. [9], assuming that the amount of secondary cases increases exponentially and the constant doubling time (*td*) is related with *R*_0_ (new infections per generation) and the infectious period (*T*) as:(5)$$R_{0}=1+\left(\frac{T}{td}\right)\times log_{e}2$$

Assuming an infectious period of seven days and ordering the PCR-positive wild boar cases by suspicion data, the average time necessary to double the number of cases was computed as formulated by Marcon et al. [29] as 39 days.

#### 2.5.2. Method 2: Estimating *R*_0_ from the Force of Infection (λ)

Assuming that the age distribution of a specific population is exponential and the mortality rate can be assumed to be constant for all age classes (as in hunted wild boar), the *R*_0_ parameter by can be computed based on the force of infection [26,42,43,44]:(6)$$R_{0}\approx 1+\left(L\times \lambda \right)$$
where *L* is the average life expectancy (30 months) and the previously estimated maximum value of *λ* was applied.

#### 2.5.3. Method 3: Estimating *R*_0_ from the Proportion of Infected

Considering the numbers of individuals who were susceptible at the start (*S*_0_) and end (*S_f_*) of the epidemic period (2011–2012 and 2012–2013 hunting seasons), *N* is the total population of ASF-tested animals and *C* is the number of cases (i.e., individuals who have been infected, seropositives), the *R*_0_ parameterization proposed by Becker [45] was applied:(7)R0=N−1C×ln{S0+12Sf−12}
where the number of susceptible individuals *S*_0_ is defined by Equation (4) and *S_f_* is defined as *S*_0_ minus the number of cases that occurred in 2011–2013 (= 4). The standard errors and 95% confidence intervals (95% CI) of the certainty of the last estimated *R*_0_ were calculated as follow:(8)SE(R0,e)=N−1C∑j=Sf+1S01j2+CR0,e2(N−1)2

#### 2.5.4. Method 4: Estimating *R*_0_ from a Simple Susceptible-Exposed-Infectious-Recovered (SEIR) Model (Optimization Analysis)

The typical susceptible-exposed-infectious-recovered (SEIR) model, largely used for the so-called “immunizing infections”, has been applied to validate the *R*_0_ estimations [28]. Assuming that the population is completely susceptible at the beginning of the epidemic, the model is described by the system of differential equations based on time (*t*):(9)$$\frac{dS\left(t\right)}{dt}=-\beta S\left(t\right)I\left(t\right)$$
(10)$$\frac{dE\left(t\right)}{dt}=\beta S\left(t\right)I\left(t\right)-fE\left(t\right)$$
(11)$$\frac{dI\left(t\right)}{dt}=fE\left(t\right)-rI\left(t\right)$$
(12)$$\frac{dR\left(t\right)}{dt}=rI\left(t\right)$$
where animals are classified as susceptible *S*(*t*), exposed *E*(*t*), infectious *I*(*t*), or recovered *R*(*t*), and *β* represents the rate at which two specific individuals come into effective contact per unit time. Considering the estimated values of *R*_0_ to be validated, *β* was input to the final model as:(13)$$\beta =\frac{R_{0}}{N\times D}$$
where *N* is the population size and *D* is the duration of infectiousness (5–7 days). The rate at which pre-infectious animals become infectious is defined as *f* in the differential equations (equal to 1/average pre-infectious period) and *r* is the rate at which individuals recover from being infectious (equal to 1/infectious period) [46,47,48,49]. Specifically, we fit the cumulative number of cases estimated by the model to the cumulative number of case notifications at time *t*. Confidence intervals for *R*_0_ estimates were constructed by generating sets of realizations of the best-fit curve using parametric bootstrap [50]. The best estimation was evaluated based on the log-likelihood deviance [51].

The simulation was implemented starting from the first virus-positive detection in wild boar (22 January 2012) with day as the time unit, for a total of 1500 time points. At the start point, the susceptible population was assumed to be 13,550 wild boars, with one virus-positive (I) and no seropositive animals. To detect the most accurate *R*_0_ value for representing the observed data (total number of infected and susceptible, peak time, and last virus reporting), a Monte Carlo simulation with 50,000 iterations after a burn-in for convergence of 10,000 observations was performed to find the optimal parameter. A scatter plot comparison was applied, according to Saltelli et al. [52].

### 2.6. Effective Reproduction Number (R_e_) Estimation

As *R_e_* defines the number of subsequent cases at a specific time, to evaluate the current situation in Anglona-Gallura, the serological data from the last three hunting seasons (2017–2020) were used to calculate *R_e_* based on the typical formulation proposed by Diekmann et al. [53] and confirmed by several authors. Defining *s* as the proportion of the population that is susceptible at time *t* of the epidemic, the estimate of *R_e_* can be computed as:(14)$$R_{e}=R_{0}\times s_{t}$$

Four different calculations of *R_e_* were computed based on the different *R*_0_ parameterizations, defining *s_t_* as 1–*z_f_*.

*R_e_* was optimized using a Bayesian inference of the stochastic SEIR model. This formulation considers that the probabilistic nature of the contagion assumes a negative binomial distribution. Thus, starting from the real-data derivate by periodic reports at time *t*, the change in the cumulative number of cases can be defined as: ∆C(t)=C(t)−C(t−τ), where *τ* denotes the time interval between reports, which is equal to one year in this dataset. Consequently, via Bayes’ theorem, the distribution of the number of future cases can be predicted by applying the probabilistic formulation of a model based on the value of *R_e_* and ∆*C*(*t*) as:(15)$$P\left[R_{e}|∆C\left(t+\tau \right)\leftarrow ∆C\left(t\right)\right]=\frac{P\left[∆C\left(t-\tau \right)\leftarrow ∆C\left(t\right)|R_{e}\right]*P\left[R_{e}\right]}{P\left[∆C\left(t-\tau \right)\leftarrow ∆C\left(t\right)\right]}$$
where *P*[*R_e_*] is the prior distribution of *R_e_*, the denominator is a normalization factor, and the notification of new cases in the next reports allows the estimation of the probability distribution function of *R_e_* as the posterior [54,55,56]. The model was iterated several times using the posterior distribution at time *t* (i.e., the probability *R_e_* distribution from previous reports) as the prior for new cases at *t* + *τ*. Uncertainty bounds (Bayesian credible intervals, BCI) were obtained by extracting the average and maximum-likelihood values of *R_e_* by its estimated distribution.

All parameter computations were solved using *MATLAB software* (version 7.10.0, R2010a, The MathWorks Inc., Natick, MA, USA). The SEIR models based on the different *R*_0_ parametrizations were stochastically implemented in the software package *Berkeley Madonna* (Version 8.3.18, Regents, University of California, CA, USA). The Bayesian stochastic SEIR model was implemented using *R software* ‘BRugs’ (Version 3.6.2, R-Foundation for Statistical Computing, Vienna, Austria) and *Open BUGS* (version 3.2.3).

## 3. Results

Based on the serological data of the first two hunting seasons described in Table 1, the daily force of infection (*λ*) estimated by the average contact rate was 0.0056 (95% CI = 0.0016–0.0092; Table 3). The catalytic model estimation was about 0.0024 (95% CI = 0.0013–0.0035), with a low deviance between observed and predicted data (goodness of fit) of 56.85, indicating the good fitting of the model. The parametric calculation based on *A* estimated a daily force of infection of 0.0012 (95% CI = 0.0004–0.0021). As illustrated in Figure 4a, the number of infected animals during the initial study period (2011–2013) increased exponentially, with a doubling time (*td*) of 39 days. Using this doubling time, we estimated *R*_0_ to be 1.124 (95% CI = 1.103–1.145) during the first two hunting seasons. As reported in Table 3, the number of secondary cases *R*_0_ calculated based on *λ* was described by a value of 1.165 (95% CI = 1.027–1.187). Based on the proportion of infected, an *R*_0_ of 1.170 (95% CI = 1.009–1.332) was estimated for the first two years under study. The estimates of the reproduction number obtained by the three methods were found to be consistent with each other (in the range R ≈ 1–1.3, with overlapping CIs). The estimation method applied to the *R_e_* calculation for the last three years (2017–2020) showed a value of 0.802 (95% CI = 0.612–0.983).

Considering the total wild boar population (13,550) at the start simulation point (22 January 2012), based on the different parameterizations of *R*_0_ ASF-specific for the WB-MU of Anglona-Gallura, four different curves describing the total number of infected animals were estimated by the optimization method with the SEIR models (Figure 6). The first model fitted using a value of *R*_0_ of 1.124 described an infectious curve with an expected peak of 53 virus-infected wild boar at time point 729 (20 January 2014) and estimated the last peak at time 1315 (29 August 2015) (green line). The second SEIR fitted by the application of the *λ* value in the *R*_0_ calculation (*R*_0_ = 1.165) estimated a peak of 88 virus-positives at time 619 (2 October 2013) and the last at time point 1112 (7 February 2015) (blue line). The third system of differential equations implemented with an *R*_0_ of 1.170, obtained by the number of susceptible animals, predicted a total of 93 infected animals at time 610 (23 September 2013) and the last virus-positive individual at time 1091 (17 January 2014; blue line).

A total of 164 cases (25 PCR-positive animals and 139 antibody-positive animals) occurred in Anglona-Gallura with an epidemic peak date at time point 657 (9 November 2013), and the last infected animal detection occurred at time 1434 (25 December 2015). The best fitting value of *R*_0_ obtained using the SEIR optimized method was ≈ 1.139 (95% CI = 1.123–1.153), with a peak at time point 687 (9 December 2013) with 64 infected animals, and the last positive animal detected at time point 1236 (11 June 2015) (black solid and dashed lines). Considering the scatter plot and log-likelihood deviation, the values of *R*_0_ estimated by the third method (based on susceptible animals) seems to more accurately describe the observed data with an overall deviance of 132 (10 degrees of freedom). The Bayesian SEIR model estimated the overall average of the effective reproduction number as 0.923 (BCI = 0.812–1.033) with level of uncertainty between observed and estimated (maximum likelihood) measured by the width of the BCI, which decreased as case observations increased (Figure 7). At later times, *R_e_* < 1 (effective reproduction number = 0.769, BCI = 0.660–0.879) as a result of averaging periods in which the epidemic declined.

## 4. Discussion

Despite its importance, to date, few epidemiological methods have been proposed to determine the end of an epidemic and the ASF-free status of a territory [18,20,57,58]. In this study, we applied different mathematical approaches to estimate the main epidemiological parameters that are able to describe the evolution and the current epidemic status of ASF in northern Sardinia. Furthermore, a context-specific estimation of the basic reproduction number can be used to calculate the herd immunity threshold, which is the minimum percentage of individuals in the population that would need to be vaccinated to ensure a disease does not persist. The last virus detection within this area was December 2015, although seropositive wild boars are still being detected during active surveillance. Our estimates for *R*_0_ at first virus detection (PCR-positive) in wild boar ranged from 1.24 to 1.70, whereas the *R_e_* estimation of 0.802 suggests that the disease is now under control.

Even though the estimation of the main epidemiological parameters and particularly the effective reproduction number is fundamental to understanding the current status of an epidemic, studies focused on this aim are lacking in the literature [9,18,27,28,29,59]. These estimations strictly depend on species (i.e., wild or domestic pigs), ASFV genotype, and type of data available (i.e., experimental or field data). Thus, the *λ* and *R*_0_ values proposed in the literature for ASF vary widely and should be considered as context specific.

Several gaps in the ASF epidemiology still remain, generating additional uncertainty when defining the fade out of a virus in the wild boar population; the undefined role of surviving seropositive animals and their possible role as carriers are two of the most influential factors [60,61,62,63]. Several epidemiological gaps are related to determining which factors could play a crucial role in disease trend in specific areas or countries, rather than others. As hypothesized in previous studies, differences in ASF epidemic curves could be mainly connected with disease management, the awareness of farmers, the presence of illegal, free-ranging pigs, and wild boar density [29,30,57]. The main limit of this study is the non-uniform sampling and the lack of samples from passive surveillance.

Although hunting is equally managed inside the IZ, as described by Cappai et al. [30], not all the municipalities of WB-MU Anglona-Gallura are included; consequently, different control measures are applied. As most of the samples were obtained from hunting season (three months/year), the dataset used was not free from sampling bias. Although the mathematical formulations applied are considered the best possible for the available data, the *λ* parameters could be slightly underestimated, particularly those proposed by Methods 2 and 3. However, the large amount of data collected at least partially limited these biases.

## 5. Conclusions

Our findings confirmed that among the different ASF cycles [8], in Sardinia, the wild boar populations play an ancillary epidemiological role in ASF maintenance, in contrast to most of the European countries [29,33,63,64]. The mathematical estimates of *R*_0_ and *R*_e_ revealed that at the end of the PCR-positive detection period, *R_e_* progressed to and remained below zero and the infection faded out spontaneously despite the presence of seropositive wild boars. However, even if the mathematical values of the epidemiological parameters indicate that the virus is fading out, the conclusions have to be further confirmed:(1)In any European area, the sampling intensity achievable through recreational hunting will always be too low to detect the virus at a very low prevalence in localized pockets of infection [65,66];(2)The efficiency of passive surveillance proves that the absence of the virus (i.e., number of negative carcasses found with respect to the local wild boar population) is far from being known and standardized;(3)The absence of virus-positive animals among the hunted wild boar for a long time, as associated with serological data, provides valuable information for confirming that the disease is moving toward self-extinction; and(4)The seropositive and virus-negative animals that are still considered carriers despite their presence do not lead to any further virus detection, as confirmed by this study and in other similar epidemiological situation (e.g., in part of Estonia and Latvia). Notably, ASFV-specific antibodies in a wild boar population were found several years after the fading out of the virus [20,60,61,62,63].

In this epidemiological landscape, hopefully observed for Euro-Asiatic ASF in the near future, it is necessary to further study and identify a specific technical approach that is able to merge epidemiological analyses, field data, and scientific evidence to minimize the risk of missing possible persistence of ASF at a very low prevalence and to ensure that restrictions are not maintained for too long, given low numbers of detected seropositive wild boars.

## Figures and Tables

**Figure 1 vaccines-08-00521-f001:**
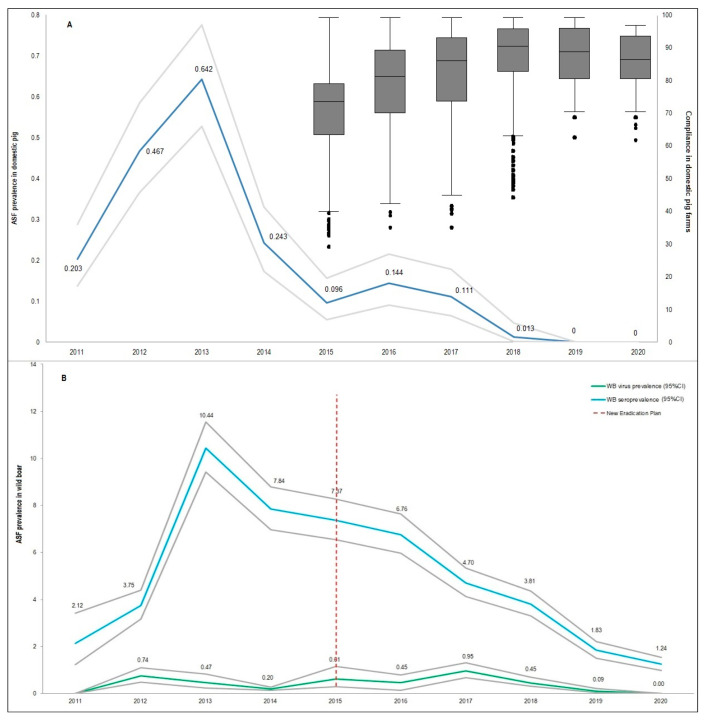
(**A**) Sardinian African swine fever (ASF) trend in domestic pigs is described as estimation (blue line) and 95% confidence interval (CI; gray lines), for the years 2011–2020. The secondary axis measures the percentage of compliance in domestic pig farms (with the rules of ASF-EP15-18), represented as box plots. The whiskers represent the minimum and maximum values and the ends of the rectangle represent the first and third quartiles, whereas the segment inside the rectangle is the median value and the dots represent the outlier values. The last box plot related to 2020 refers to the first six months of the year. (**B**) Sardinian ASF disease prevalence trend in wild boar. The virus-positive trend is represented by the green line and 95% CI (gray lines). The seropositive trend is represented by the blue line and 95% CI (gray lines). Data are expressed as percentage.

**Figure 2 vaccines-08-00521-f002:**
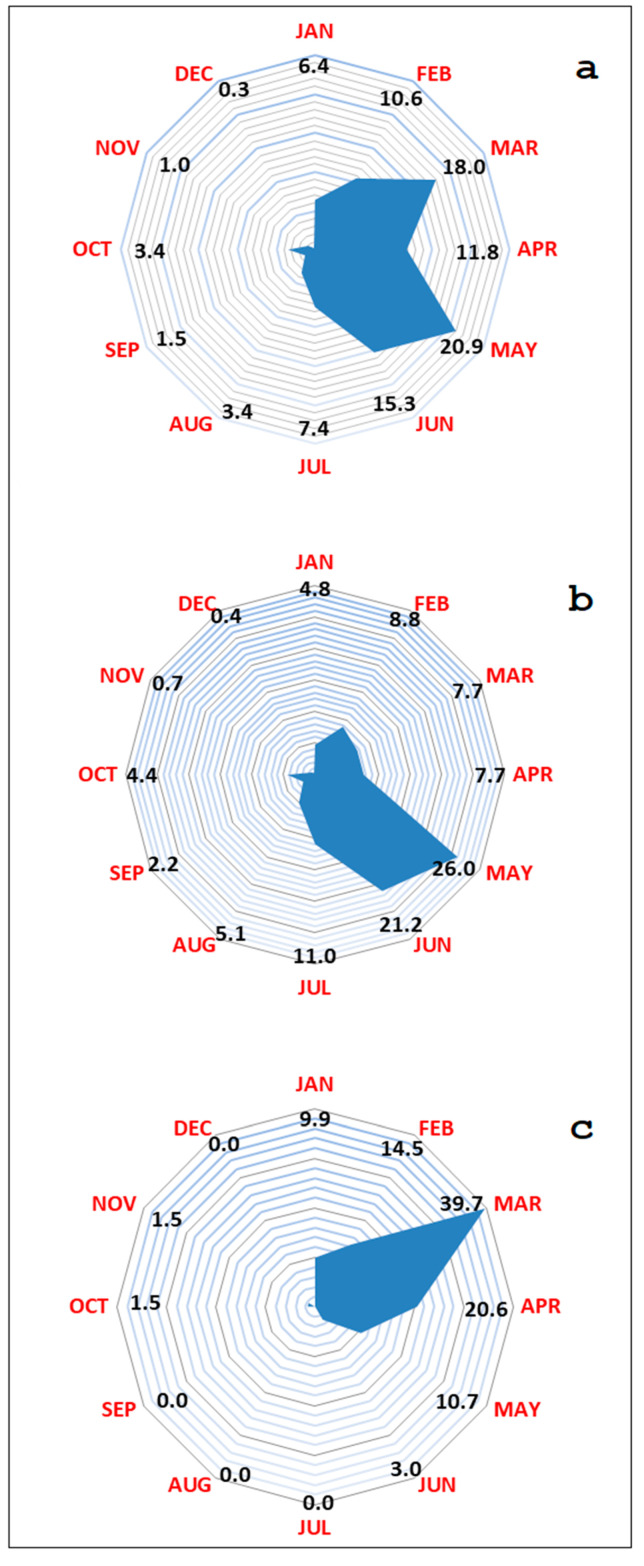
(**a**) Overall seasonality of African swine fever events in the Anglona area of the Sardinian region (**b**) in domestic pig farms and (**c**) in the wild boar population, from January 2012 to December 2019 corrected for number of observation months.

**Figure 3 vaccines-08-00521-f003:**
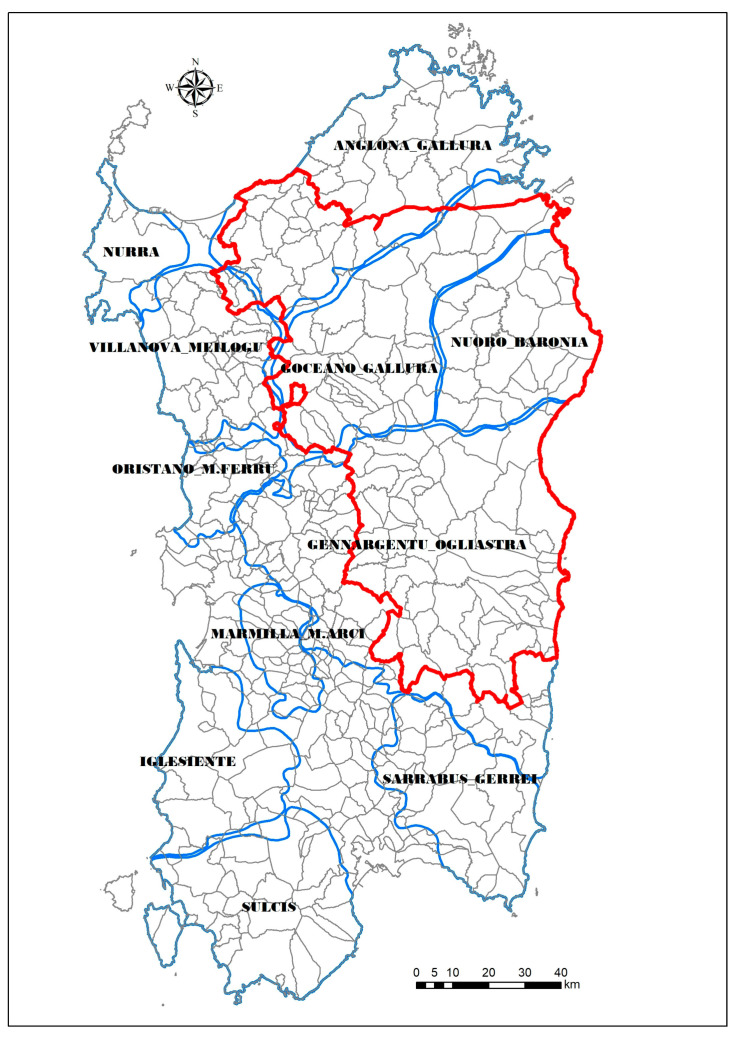
Contextualization of the Sardinian data based on wild boar management unit. The wild boar infection zone (IZ) is indicated by the red line.

**Figure 4 vaccines-08-00521-f004:**
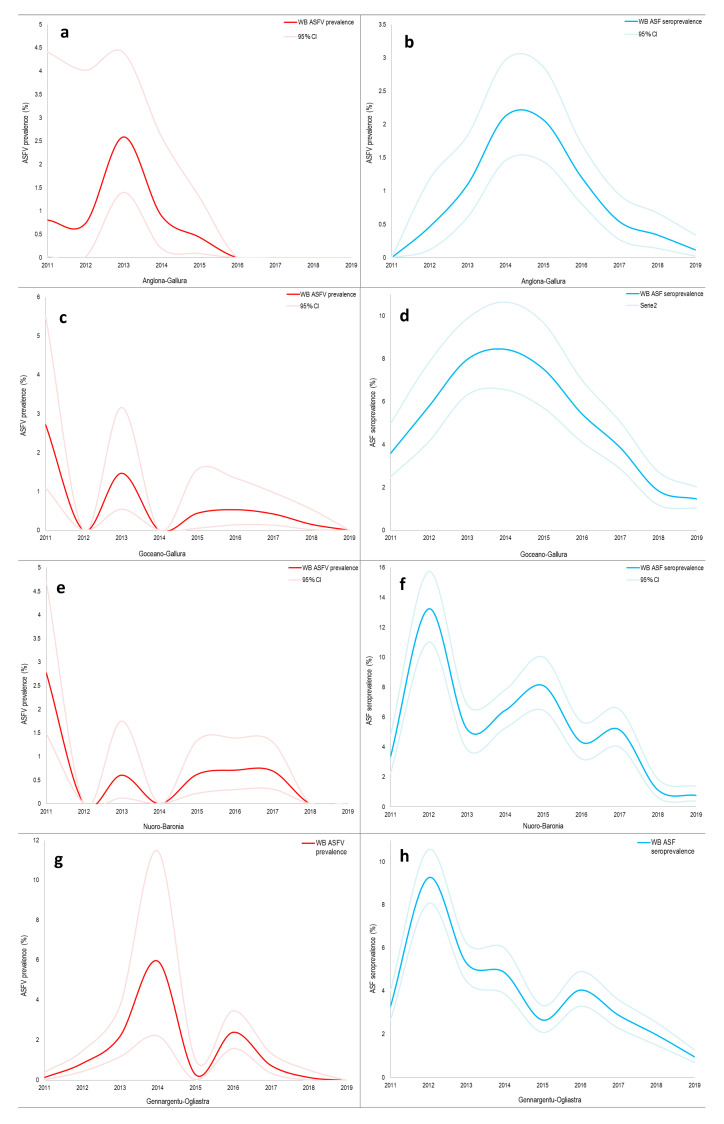
(**a**–**h**) ASF virus prevalence trend in wild boar based on 2011–2020 data of the four wild boar management units included in the infected zone. The trends are represented as African swine fever virus and seroprevalence trends for the wild boar management units of (**a**,**b**) Anglona-Gallura, (**c**,**d**) Goceano-Gallura, (**e**,**f**) Nuoro-Baronia, and (**g**,**h**) Gennargentu-Ogliastra. Data are expressed as percentage of prevalence.

**Figure 5 vaccines-08-00521-f005:**
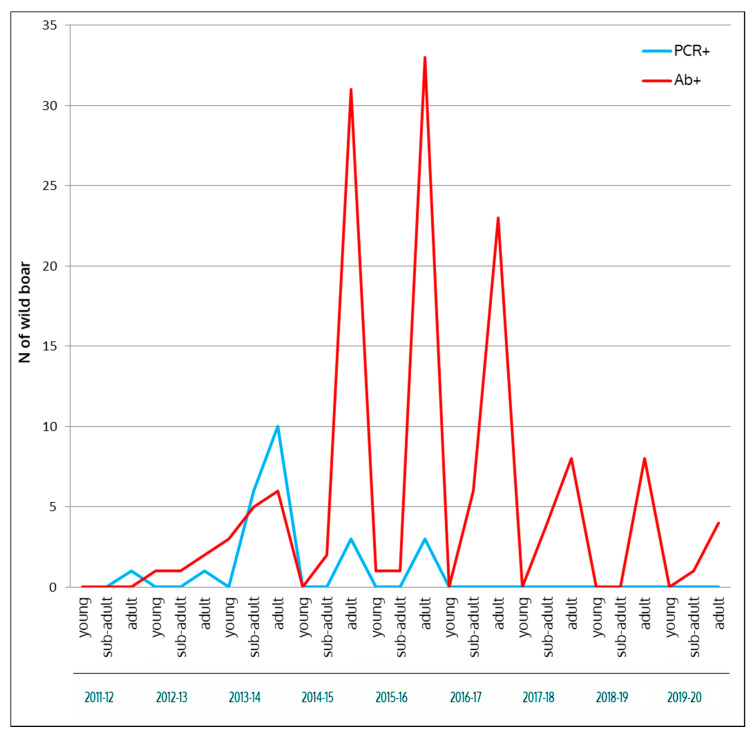
Number of ASF virus (ASFV) PCR-positive and seropositive animals detected in Anglona-Gallura during 2011–2020, divided into young, subadult, and adult categories.

**Figure 6 vaccines-08-00521-f006:**
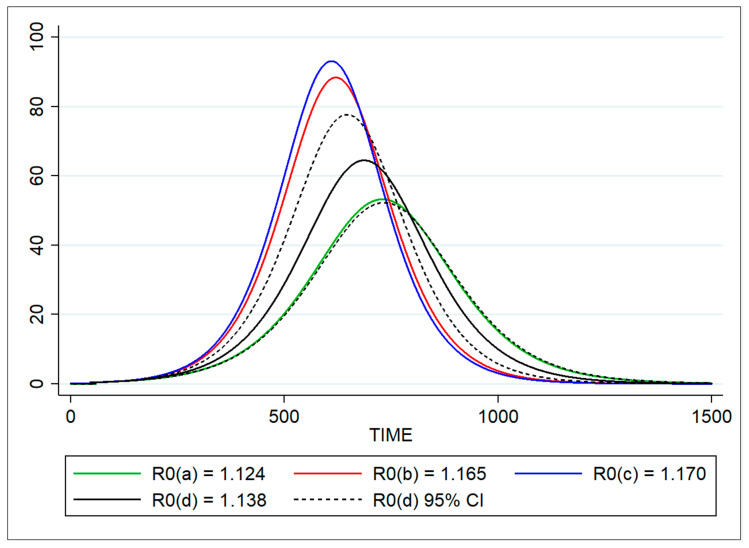
Number of ASFV infectious animals estimated in Anglona-Gallura by the four SEIR models fitted for 1500 time points. The blue line represents the curve of infectious individuals estimated by the *R*_0_(*a*) based on doubling time (red line), *R*_0_(*b*) based on force of infection (green line), *R*_0_(*c*) based on the proportion of infected (blue line), and *R*_0_(*d*) based on SEIR model (optimization). The dashed lines represent the 95% CI of the optimized *R*_0_.

**Figure 7 vaccines-08-00521-f007:**
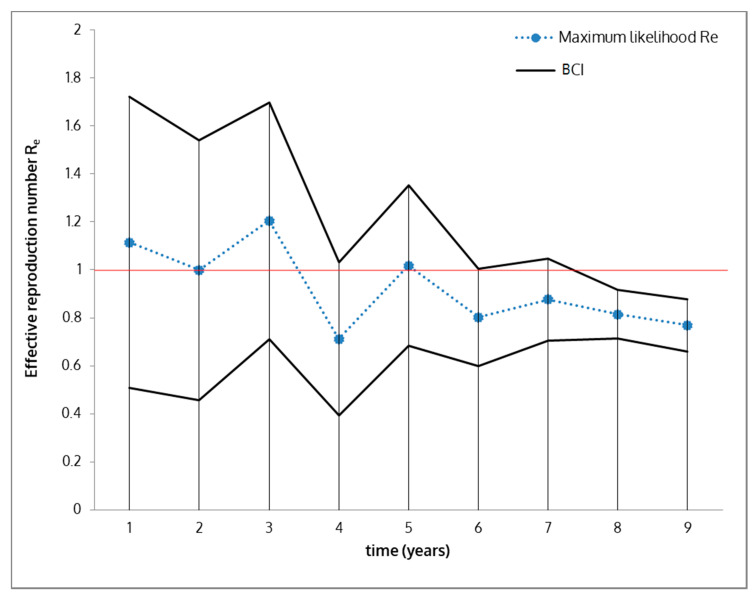
Sequential Bayesian estimation of the full distribution of effective reproduction number (R_e_) leads to the estimation of its maximum-likelihood value (blue dots) and BCIs (black lines). Upper limit of the estimate is below 1, indicating virus fade out.

**Table 1 vaccines-08-00521-t001:** Summary of baseline database created for the active surveillance in Anglona-Gallura area, including time of hunting season (from 2011 to 2020), wild boar age classes (young, subadult, and adult), the number of wild boar tested for ASF virus presence or antibody presence, and the virus prevalence or seroprevalence. Data are presented as number and percentage, or number and prevalence with 95% confidence intervals.

Hunting Season	Wild Boar Age Classes	Wild Boar Virologically ^1^ Tested	Virus Prevalence	Wild Boar Serologically ^2^ Tested	Seroprevalence ^3^
		n (%)	(95% CI)	n (%)	(95% CI)
2011–2012	Young	40 (33.6)	0 (0–0)	272 (28.1)	0 (0–0)
	Subadult	47 (39.5)	0 (0–0)	277 (28.6)	0 (0–0)
	Adult	32 (26.9)	6.4 (0.1–16.2)	418 (43.3)	0 (0–0)
	**Total**	**119**	**0.8 (0.2–4.6)**	**967**	**0 (0–0)**
2012–2013	Young	21 (17.2)	0 (0–0)	131 (19.5)	0.8 (0.0–4.2)
	Subadult	34 (27.9)	0 (0–0)	171 (25.5)	0.6 (0.0–3.2)
	Adult	67 (54.9)	1.5 (0.0–8.0)	369 (55.0)	0.5 (0.1–1.9)
	**Total**	**122**	**0.8 (0.0–4.5)**	**671**	**0.6 (0.2–1.5)**
2013–2014	Young	33 (6.8)	0 (0–0)	55 (4.5)	5.5 (1.1–15.1)
	Subadult	191 (39.2)	3.1 (1.2–6.7)	368 (30.3)	1.4 (0.4–3.1)
	Adult	263 (54.0)	4.2 (2.1–7.3)	792 (65.2)	0.8 (0.3–1.6)
	**Total**	**487**	**3.5 (2.0–5.5)**	**1215**	**1.1 (0.6–1.9)**
2014–2015	Young	33 (10.1)	0 (0–0)	80 (5.3)	0 (0–0)
	Subadult	98 (29.9)	0 (0–0)	410 (27.3)	0.5 (0.1–1.7)
	Adult	197 (60.0)	1.5 (0.3–4.4)	1014 (67.4)	3.5 (2.0–4.2)
	**Total**	**328**	**0.9 (0.2–2.6)**	**1504**	**2.1 (1.4–3.0)**
2015–2016	Young	53 (7.8)	0 (0–0)	95 (5.8)	1.1 (0.0–5.7)
	Subadult	182 (27.0)	0 (0–0)	420 (25.4)	0.2 (0.0–1.3)
	Adult	440 (65.2)	0.7 (0.1–2.0)	1137 (68.8)	2.9 (2.0–4.0)
	**Total**	**675**	**0.4 (0.0–1.3)**	**1652**	**2.1 (1.5–2.9)**
2016–2017	Young	64 (7.1)	0 (0–0)	117 (4.9)	0 (0–0)
	Subadult	244 (27.1)	0 (0–0)	657 (27.7)	0.9 (0.3–2.0)
	Adult	591 (65.8)	0 (0–0)	1600 (67.4)	1.4 )0.9–2.1)
	**Total**	**899**	**0 (0, 0–0)**	**2374**	**1.2 (0.8–1.7)**
2017–2018	Young	71 (6.1)	0 (0–0)	132 (5.9)	0 (0–0)
	Subadult	364 (31.3)	0 (0–0)	679 (30.6)	0.6 (0.2–1.5)
	Adult	729 (62.6)	0 (0–0)	1408 (63.5)	0.6 (0.2–1.1)
	**Total**	**1164**	**0 (0, 0–0)**	**2219**	**0.5 (0.3–0.9)**
2018–2019	Young	78 (6.1)	0 (0–0)	97 (4.2)	0 (0–0)
	Subadult	290 (22.8)	0 (0–0)	531 (22.7)	0 (0–0)
	Adult	901 (71.1)	0 (0–0)	1709 (73.1)	0.5 (0.2–0.9)
	**Total**	**1269**	**0 (0, 0–0)**	**2337**	**0.3 (0.1–0.7)**
2019–2020	Young	53 (4.0)	0 (0–0)	92 (3.9)	0 (0–0)
	Subadult	339 (25.8)	0 (0–0)	599 (25.3)	0.2 (0.0–0.9)
	Adult	921 (70.2)	0 (0–0)	1674 (70.8)	0.2 (0.1–0.6)
	**Total**	**1313**	**0 (0, 0–0)**	**2365**	**0.2 (0.1–0.5)**

^1^ Virus presence was assessed by real-time PCR, quantitative PCR, or Malmquist test. ^2^ Wild boar serologically tested were those tested with at least the screening ELISA test (Ingezim PPA Compac^®^, Ingenasa, Madrid, Spain), and eventually confirmed (if positive) with immunoblotting (IB), in accordance with the Manual of Diagnostic Tests and Vaccines for Terrestrial Animals [15]. ^3^ Serum samples were considered positive when they scored positive in both the screening (ELISA) and the confirmatory (IB) tests.

**Table 2 vaccines-08-00521-t002:** Estimate of standard error of the mean (SEM), 95% CIs of the mean, and width of CI of the original data observed during the Sardinian hunting season.

Variable	N	Mean	SEM	95% CI of Mean	Width of the CI
Week 1a	23	39.874	0.859	38.189–41.558	3.368
Week 1b	21	8.056	0.350	7.369–8.742	1.373
Week 2a	22	42.073	0.943	40.223–43.922	3.699
Week 2b	24	10.632	0.448	9.753–11.511	1.758
Week 3a	21	40.000	0.878	38.278–41.722	3.444
Week 3b	22	9.476	0.391	8.708–10.243	1.535
Week 4a	24	38.000	0.973	36.092–39.908	3.816
Week 4b	22	11.000	0.471	10.076–11.924	1.848

**Table 3 vaccines-08-00521-t003:** Summary of the parameters estimated using the different methods, expressed as estimates and 95% confidence intervals (95% CIs).

Parameter	Definition	Method	Estimate	95% CI
*λ*	Force of infection (daily)	Based on contact rate *β*	0.0056	0.0016–0.0092
		Catalytic model	0.0024	0.0013–0.0035
		Based on average age at infection *A*	0.0012	0.0004–0.0021
*A*	Average age at infection (monthly)	Life expectancy	28.5	28.1–29.9
*s*	Proportion of susceptible/receptive population	Serology by age	0.952	0.937–0.991
*R* _0_	Basic reproduction number	Doubling time	1.124	1.103–1.145
		Based on force of infection *λ*	1.165	1.027–1.301
		Proportion of infected	1.170	1.009–1.332
		SEIR model (optimization)	1.139	1.123–1.153
*R_e_*	Effective reproduction number	Average value based on basic reproduction number *R*_0_	0.802	0.612–0.992
		Bayesian SEIR model (optimization) (BCI)	0.923	0.812–1.033
*f*	Infection rate	1/average pre-infectious period [46,47,48]	0.28	–
*r*	Recovery rate	1/infectious period [46,47,48]	0.15	–

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
