# Peer review of "Mathematical Approach to Estimating the Main Epidemiological Parameters of African Swine Fever in Wild Boar"

_vaccines, 2020, doi:10.3390/vaccines8030521_

Round 1
Reviewer 1 Report
In this study the authors attempt to estimate the basic and effective reproduction rates for African swine fever virus in Sardinian boar, feral swine and domestic swine, but with a particular emphasis on the wild boar data.
I am not in a position to critically assess the appropriateness of the epidemiological techniques in this study, so I will have to defer to the other reviewer on that aspect, but from the perspective of a molecular virologist, the study appears to justify its assumptions for the most part, and follows logically from the data. I think a strength of the study is that several methods of estimating these R values were reported, and gave generally similar results.
My biggest concern is with fit for this particular journal. The concepts described here are not directly related to vaccine research, but could potentially and indirectly inform research if a vaccine existed, which it does not.
My second biggest concern in reading the study is in line 315, to explain or reference the reasoning behind the estimated starting population size for wild boar, and whether this is based only on the endemic area, or all of the island.
Minor changes
20 Severe
21 “Unstoppable” seems to be the wrong word, given the rest of the abstract which shows that it is in fact controllable, and the R values calculated between 0.8 and 1.2 – please rephrase or delete this sentence as inaccurate.
89-90 Please specify details or delete this sentence instead of writing the rather vague “results never obtained before”
Table 1 – the “n” should be outside the parentheses in the third column, as it is in the fifth column.
406 – seropositive
It's quite clear overall, but the language needs a bit of polishing with help from an editor.
Author Response
Dear reviewer,
first of all we would to really thank you for the fast and precise revision of the manuscript. The matter of the effective R calculation is not of simple solution, considering that the data on active and passive surveillance of wild boar are never perfect data and considering the main concerns related to ASF virus detection. We modified all the manuscript following your suggestions and those of the other reviewers.
1. My biggest concern is with fit for this particular journal. The concepts described here are not directly related to vaccine research, but could potentially and indirectly inform research if a vaccine existed, which it does not.
Resp.1
Dear reviewer, this manuscript has been submitted to the special issue of vaccines “African Swine Fever Virus Prevention and Control”. In our opinion, the definition of the main epidemiological parameters are the first (and the most important) step for each infectious disease prevention and control. Furthermore, the R0 can be used to calculate the herd immunity threshold, which is the minimum percentage of people in the population that would need to be vaccinated to ensure a disease does not persist in the population. However, considering that your concern could also arise in the readers, in discussion we added a sentence as: “Furthermore, a context specific estimation of the basic reproduction number can be used to calculate the herd immunity threshold, which is the minimum percentage of individuals in the population that would need to be vaccinated to ensure a disease does not persist.”
2. My second biggest concern in reading the study is in line 315, to explain or reference the reasoning behind the estimated starting population size for wild boar, and whether this is based only on the endemic area, or all of the island.
Resp 2.
Thank you for this specification. We modified the sentence we specifying that the estimation is referred only to Anglona-Gallura area and adding two references of previous works where the same authors described in details the estimation procedure, based on several factors (particularly the type of land).
Minor changes
20 Severe.
The word has been corrected
21 “Unstoppable” seems to be the wrong word, given the rest of the abstract which shows that it is in fact controllable, and the R values calculated between 0.8 and 1.2 – please rephrase or delete this sentence as inaccurate.
The word has been deleted
89-90 Please specify details or delete this sentence instead of writing the rather vague “results never obtained before”
The previous sentence has been substituted with as follow: “Efficient depopulation actions against free-ranging pigs have been put in place and about 4,500 illegal pigs were culled between December 2017 and February 2020”
Table 1 – the “n” should be outside the parentheses in the third column, as it is in the fifth column.
The heading has been corrected
406 – seropositive
the word has been corrected
It's quite clear overall, but the language needs a bit of polishing with help from an editor.
- The manuscript has been completely reviewed by MDPI editing translation service
Reviewer 2 Report
Comments to the Authors of manuscript number: vaccines-934038 entitled “Mathematical approach to estimate the main epidemiological parameters of African swine fever in wild boar”.
The authors used mathematical tools to estimate the main epidemiological parameters of ASF in wild boar, which play important role in the epidemiology of ASF in Sardinia. For this purpose, correct tests and statistical tools were used. The authors clearly explained each aspect studied.
In my opinion, the manuscript is written correctly and very logically. This is especially difficult when the manuscript covers a wide period of time (2011-2020). Moreover, as Authors wrote data about ASF outbreaks in wild boar were retrieved from the Italian National Informative System for Animal Disease Notification. I have no reason to doubt the given data from domestic pig farms and wild boar population.
Authors present calculation of the risk of appearance of new infection using 3 methods and four different methods are applied for calculation of the basic reproduction number, all methods are proper.
However, there are two points, which should be corrected.
- Authors' affiliations should be in English.
- Figure 1. There is no explanation what is the meaning of dots.
Author Response
However, there are two points, which should be corrected.
- Authors' affiliations should be in English.
- Figure 1. There is no explanation what is the meaning of dots.
Dear Reviewer,
Our sincerely thanks for your fast and precise review. Following your suggestions, this new version of the manuscript has been modified adding the author’s affiliation in English language and the heading of the figure 1 has been updated adding the explanation of the dots as outliers.